# A molecular atlas of innate immunity to adjuvanted and live attenuated vaccines, in mice

Audrey Lee[1,8], Madeleine K. D. Scott[1,2,8], Florian Wimmers[1], Prabhu S. Arunachalam [1], Wei Luo [3], Christopher B. Fox[4], Mark Tomai[5], Purvesh Khatri[1,2✉] & Bali Pulendran [1,6,7✉]

Adjuvants hold great potential in enhancing vaccine efficacy, making the understanding and improving of adjuvants critical goals in vaccinology. The TLR7/8 agonist, 3M-052, induces long-lived humoral immunity in non-human primates and is currently being evaluated in human clinical trials. However, the innate mechanisms of 3M-052 have not been fully characterized. Here, we perform flow cytometry, single cell RNA-seq and ATAC-seq to profile the kinetics, transcriptomics and epigenomics of innate immune cells in murine draining lymph nodes following 3M-052-Alum/Ovalbumin immunization. We find that 3M-052-Alum/OVA induces a robust antiviral and interferon gene program, similar to the yellow fever vaccine, which is known to confer long-lasting protection. Activation of myeloid cells in dLNs persists through day 28 and single cell analysis reveals putative TF-gene regulatory programs in distinct myeloid cells and heterogeneity of monocytes. This study provides a comprehensive characterization of the transcriptomics and epigenomics of innate populations in the dLNs after vaccination.

[1] Institute for Immunity, Transplantation and Infection, Stanford University School of Medicine, Stanford University, Stanford, CA, USA. [2] Center for Biomedical Informatics, Department of Medicine, Stanford University School of Medicine, Stanford, CA, USA. [3] Department of Microbiology and Immunology, Indiana University School of Medicine, Indianapolis, IN, USA. [4] Infectious Disease Research Institute, Seattle, WA, USA. [5] 3M Corporate Research and Materials Lab, St. Paul, MN, USA. [6] Department of Pathology, Stanford University School of Medicine, Stanford University, Stanford, CA, USA. [7] Department of Microbiology and Immunology, Stanford University School of Medicine, Stanford University, Stanford, CA, USA. [8] These authors contributed equally: Audrey Lee, Madeleine K. D Scott. ✉email: pkhatri@stanford.edu; bpulend@stanford.edu

The ultimate goal of vaccination is to confer life-long protection against infection. Despite decades of research, vaccines that induce robust and durable immunity against many diseases such as malaria, seasonal influenza, and HIV remain elusive. A major hurdle to developing reliable, robust vaccines relates to the antigenicity of the vaccine components. Due to the inherent lack of antigenicity of some viral components, such as HIV envelope antigens, adjuvants are often required to enhance the robustness of the immune response[1–3]. Today, adjuvants are central components of many vaccines and are essential for maximizing the protective effect of vaccines by enhancing both the magnitude and duration of immune responses[3,4]. Recent work has focused on a new generation of adjuvants that directly target pattern recognition receptors (PRRs) on innate immune cells. Innate immune cells are key modulators of the immune system as they orchestrate adaptive immune responses and targeting specific innate pathways with adjuvants have substantially proven to fine-tune antibody and T cell responses[5–9]. Adjuvants that target TLRs, as CpG1018 (TLR9)[10–12] and MPL[13] have been included in licensed vaccines, and several others such resiquimod (R848) (TLR7/8 ligand) have demonstrated great promise in eliciting robust and durable T and B cell responses[14]. While this new generation of adjuvants has shown great promise, their mechanism of action is not well understood.

3M-052 is a novel TLR7/8 agonist and has been shown to induce potent antigen-specific immune responses in non-human primates, characterized by Th1 cellular responses as well as long-lived antibody and plasma cell responses[15–17]. Despite its potent adjuvanticity, the mechanism of action of 3M-052 remains poorly understood[16,17]. Studies examining the adjuvant potential of 3M-052 formulated in alum, or in PLGA nanoparticles, have been limited to characterizing the peripheral immune response in blood from NHPs[15–18], and little is known about the events that occur in the lymph node, which is the site where immune responses are initiated. Following vaccination, activated dendritic cells (DCs) from the local site of injection migrate to the draining lymph nodes (dLNs) present antigens to T cells, and secrete cytokines that support T and B cell activation and differentiation, thereby stimulating an immune response[19,20]. Furthermore, despite its potent activation of innate and adaptive immunity, there are at present no studies that have "benchmarked" the immune responses induced by 3M-052-adjuvanted vaccines, with those stimulated by live viral vaccines, which induce robust and durable antibody responses that can last a lifetime.

One such vaccine is the yellow fever vaccine, YF-17D, which is a live attenuated virus, and one of the most successful vaccines ever developed. Administered to more than 600 million people globally, YF-17D has an efficacy greater than 99%[21,22]. A single immunization stimulates neutralizing antibodies and robust antigen-specific CD8+ T cell responses that can confer protection that lasts several decades[23–26]. Therefore, YF-17D often serves as a gold standard model for vaccine design and benchmarking of novel vaccines. YF-17D has been shown to act through multiple receptors, including TLR2, 7, 8, 9, RIG-I, and MDA-5[24,27,28]. Furthermore, systems biological analysis of the innate and adaptive responses to YF-17D vaccination in humans has revealed transcriptional signatures in the blood, induced within a few days of vaccination that correlate with and predict the ensuing antibody and CD8+ T cell responses[24,29]. However, such analyses have been confined to the blood, and there is currently no knowledge about the transcriptional networks that are induced in the dLNs. In addition, emerging evidence suggests that infection and vaccination can induce persistent epigenetic changes in innate immune cells[30–32]. In the present study, we utilize flow cytometry, single cell RNA (scRNA-seq), and single cell ATAC sequencing (scATAC-seq) to generate a comprehensive cellular,

transcriptomic and epigenomic map of the innate immune response in the dLN at early and late timepoints after immunization with YF-17D or 3M-052-Alum adjuvanted antigen. Our analysis provides a rich high resolution data set of the epigenomic and transcriptomic map of the innate immune response in lymph nodes, in the important context of vaccination.

## Results

**Immunization with antigen plus 3M-052/Alum induces activation and recruitment of myeloid cells in the dLNs.** In mice, 3M-052-Alum/OVA induced a robust humoral response at day 14 and 28 post-immunization, as indicated by the high titer of total IgG, IgG2c, IgG3 (Supplementary Fig. 1a). Immunization with 5 μg 3M-025-Alum significantly enhanced germinal center response, T follicular helper cell response, and increased the antibody titers compared to Alum alone as adjuvant (Supplementary Fig. 1a, b).

To generate an overview of the innate response in the skin-dLNs to vaccination with OVA plus 3M-052 formulated in alum[15–17] adjuvant, we measured the kinetics and activation of distinct innate populations by flow cytometry at day 1, 3, 7, 14, and 28 post-immunization. We performed scATAC-seq and scRNA-seq at an early (day 1) and late (day 28) timepoint after subcutaneous immunization with the model antigen ovalbumin (OVA) antigen adjuvanted with 3M-052-Alum (Fig. 1a). Using flow cytometric staining, innate cell subsets in the lymph nodes were identified as the following: monocytes as CD11b+Ly6C$^{hi}$, macrophages as CD11b+Ly6C$^{lo}$F4/80+, DCs as CD11c$^{hi}$MHCII$^{hi}$ and subsets of DCs further subdivided into resident CD8+ and CD11b+, and migratory CD103+ and CD11b+ DCs (gating strategy shown in Supplementary Fig. 1c). 3M-052-Alum/OVA stimulated a robust innate immune response in the dLNs that peaked in general at 24 h post-immunization, for both the 1 and 5 μg dose. This peak immune response was characterized by an upregulation of an activation marker, CD86, on LN resident and migratory DCs, monocytes, pDCs, and macrophages (Fig. 1b). In addition, activation of myeloid cells was similar at 1 and 5 μg doses of 3M-052-Alum, but a higher frequency of monocytes, NK cells, and resident DCs could be detected in the dLNs at day 3 following immunization with 5 μg dose compared to 1 μg dose (Fig. 1c). After 24 h post-immunization, we observed a striking increase in the frequency of monocytes (>7.5-fold) and increase of migratory DCs, both CD103+ (~1.5-fold) and CD11b+ subsets (~2-fold), and NK cells (>2-fold). There was also an elevated frequency of migratory and resident CD11b+ DCs on 3 days post-immunization (Fig. 1c and Supplementary Fig. 1d). Migratory CD11b+ DCs were further gated out and this revealed an increase in CD205+CD24+ dermal DC subset on both day 1 and 3 post-immunization (Supplementary Fig. 1d). Thus, our result indicates the continuous recruitment of migratory CD11b+ DCs up till day 3.

**Immunization with antigen plus 3M-052/Alum induces a global anti-viral innate immunity program in draining LNs.** Following the observation that activation of innate immune cells in the dLNs largely peaked at day 1, we further characterized the transcriptional and epigenetic landscape of these innate cell subsets at day 1 post-immunization with 5 μg 3M-052-Alum/OVA. We isolated inguinal lymph nodes at day 1 post-immunization, performed magnetic separation to deplete the majority of T and B lymphocytes, and sorted major innate immune populations (Ly6C+ cells, DCs, pDCs). We collected a total of 21,664 cells, consisting of 3830 cells at baseline (day 0) and 9134 cells at day 1 after immunization. Louvain clustering and UMAP embedding segregated the single cells into clusters inferred based

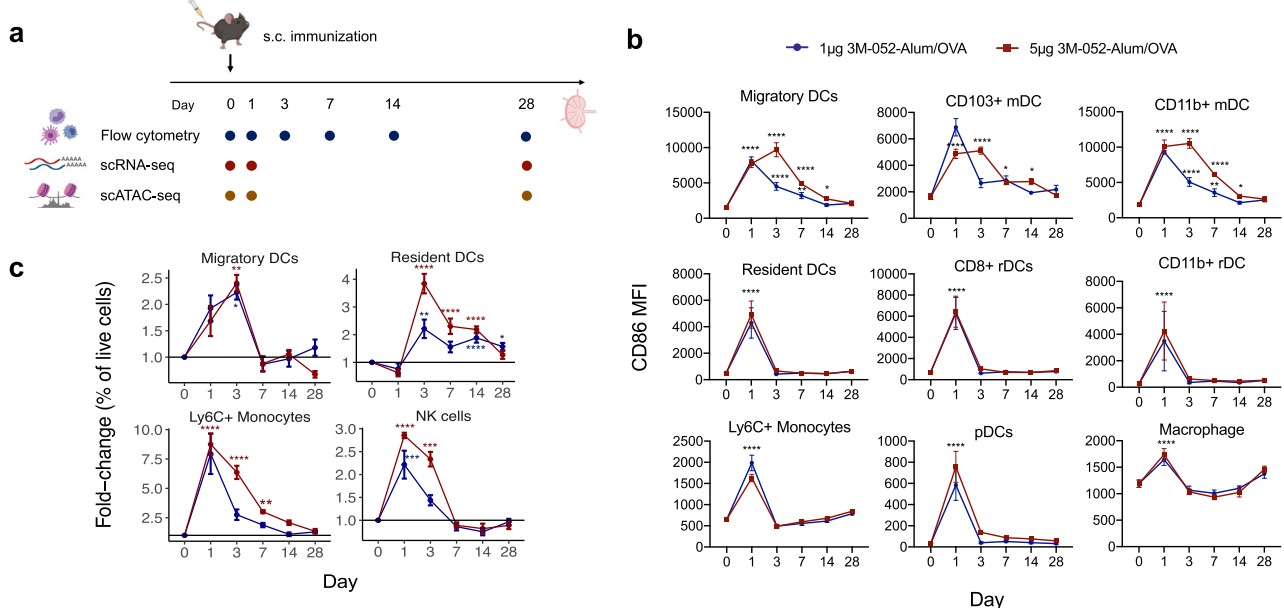

**Fig. 1 3M-052-Alum/OVA induces changes in activation and cell frequency of innate immune cells in the draining LN on Day 1. a** Schematic of flow cytometry phenotyping, CITE-seq and scATAC-seq in this study. **b** Mean ± SEM of CD86 median fluorescence intensity (MFI) in innate immune cell subsets from the dLNs at each timepoint (*n* = 5 for each group; naïve group pooled from three independent experiments, *n* = 15). Two-way ANOVA with Dunnett's multiple comparisons test in comparison to Day 0 (*, adjusted *p*-value < 0.05; **, padj < 0.01; ****, padj < 0.0001). **c** Fold-change of mean % live cells ± SEM across innate immune cell subsets in the dLNs at day 1 post-immunization compared to naïve control (*n* = 5 for each group; naïve group pooled from three independent experiments, *n* = 15). Two-way ANOVA with Dunnett's multiple comparisons test in comparison to Day 0 (*, adjusted *p*-value < 0.05; **, padj < 0.01; ****, padj < 0.0001). Source data are provided as a Source Data file.

on their gene expression patterns. The resulting clusters represented the major myeloid subsets present in the dLN, such as resting and activated monocytes, DCs (Fig. 2a). Lymphoid subsets including NK cells and ILCs, and a small population of γδ T cells, Tregs, and CD8+ T cells were also present (Fig. 2a). We identified distinct DC subsets including migratory DCs, cDC1, cDC2, pDCs, and a population of recently described transitional DCs (tDCs)[33–35] (Fig. 2b). In agreement with previous studies, tDCs exhibited high expression of genes canonically associated with pDC and cDC2, including *Irf4* and *Cx3cr1*, as well as intermediate levels of *Siglech* and *Irf8* (Sup Fig. 2a)[34]. Migratory DCs were further re-clustered to identify distinct subpopulations at a higher resolution. The subclusters could be annotated as resting and activated Langerin+ DCs and CD205+ dermal DCs based on their characteristic markers (Supplementary Fig. 2b). The activated phenotype of these migratory subpopulations in the dLNs indicates the migration of activated skin DC subsets into the dLN at day 1 post-immunization, consistent with our flow cytometry data (Supplementary Fig. 1b).

To investigate the innate immune activation in the dLNs on day 1, we analyzed differentially expressed genes between day 1 and day 0 cells within each annotated cluster. Genes with an FDR less than 0.05 and absolute log fold change above 0.25 were deemed significant. Monocytes demonstrated the largest number of differentially expressed genes (1192 DEGs) followed by tDCs (1150 DEGs), suggesting that both populations respond strongly to 3M-052-Alum/OVA 1 day after immunization (Fig. 2c).

3M-052-Alum/OVA induced a global type-I interferon and antiviral signature across all dLN cell subsets, including various innate cell populations and CD8 T cells and T regulatory cells (Fig. 2d). This type-I interferon response was characterized by the expression of interferon-stimulated genes (ISGs), such as *Ifit3*, *Irf7*, and *Isg15*. Several of these ISGs were previously found to be antiviral effector ISGs that are broadly induced across many cell types[36]. In line with the current understanding of TLR7/8

expression in mice, high expression of *Tlr7* was observed in monocytes and pDCs, with lower expression in XCR1- DC, Treg and CD8+ cell subsets. The function of murine TLR8 remains controversial, however, *Tlr8* expression could be detected solely in monocytes in the dLN[37,38] (Supplementary Fig. 2c). This suggests that 3M-052, as a TLR7/8 agonist, may primarily function through stimulation of these TLR7-expressing cell types.

In addition, myeloid cells, in particular monocytes and DC subsets, upregulated expression of genes involved in TLR signaling pathway (e.g., *MyD88*, *Cxcl9*, and *CD40*), consistent with the fact that 3M-052 stimulates TLR signaling. Other innate immune processes including complement (e.g., *C3*, *Cfb*) and inflammasome (e.g., *Casp3*, *Il18*, *Il1rn*) genes were also found to be significantly upregulated mainly in monocytes and tDCs, and to a lesser extent in XCR1− and XCR1+ DCs (Fig. 2d)[29,39]. In comparison, lymphoid cells such as innate lymphoid cells (ILCs), γδT cells, and CD8+ lymphocytes, exhibited higher expression of genes that are involved in cytokine responses (e.g., *Jak2*, *Nfkbia*) when compared to myeloid cell subsets (Fig. 2d). This may indicate indirect activation through cytokines being produced by monocytes and DCs. Furthermore, we observed a downregulation of antigen processing and presentation gene modules defined by genes such as *H2-Eb1*, *H2-Aa*, and *Cd74* in monocytes, but an upregulation of these genes in XCR1− DCs, migratory DCs, ILC2, and ILC3 (Fig. 2d). This is consistent with previous reports of MHCII gene expression in ILC2 and ILC3 as well as the involvement of these innate lymphoid subsets in antigen presentation[40]. The decrease in MHCII presentation genes also supports the notion that more inflammatory monocytes, which have a relatively low expression of MHCII genes compared to resting monocytes, were recruited to the lymph nodes on day 1 after stimulation[41].

We next performed an overrepresentation analysis of blood transcriptional modules (BTMs) using DEGs (absolute logFC > 0.25, FDR < 0.05) to identify significant patterns of activation

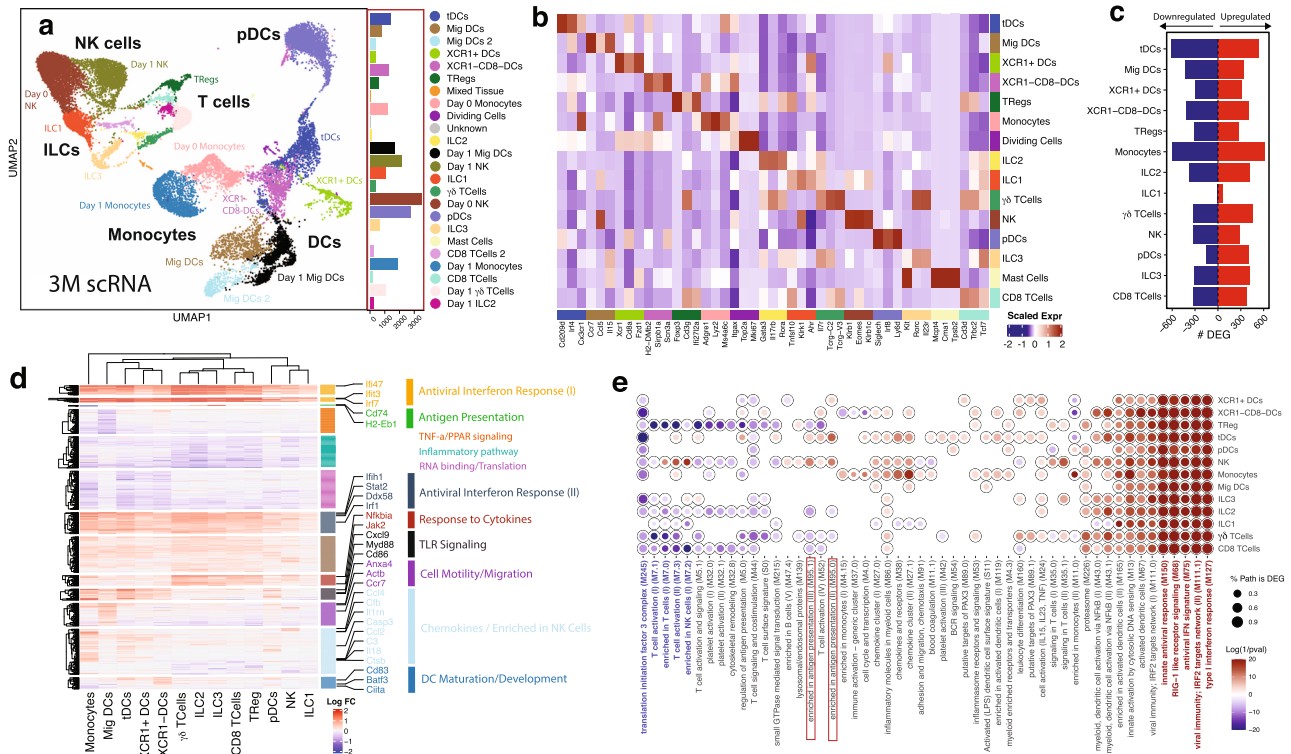

**Fig. 2 3M-052-Alum/OVA induces a global anti-viral innate immunity program in draining LN 1 day post-immunization. a** UMAP embedding of 21,664 cells classified by cell type. **b** Scaled expression of cell-type-specific genes used to classify cells by type. **c** Number of upregulated (red) and downregulated (blue) genes in each cell type, FDR < 0.05 and absolute LogFC > 0.25. **d** Log fold change of genes from (**c**) across all cell types. Genes and cell types were k-means and hierarchically clustered. Functional groups of relevant gene clusters are highlighted. **e** BTMs significantly enriched (FDR < 0.001) after 1- day post-immunization with 3M-052-Alum/OVA. Overrepresentation analysis of the genes from (**c**) was used to determine significance. Adjusted *P*-values were measured using hypergeometric distribution with Benjamini Hochberg correction.

across myeloid and lymphoid subsets[42]. This revealed modules overrepresented in tDCs on day 1 that were distinct from pDCs but shared similarities with XCR1− DCs. Specifically, similar to XCR1− DCs, tDCs exhibited an increased expression of genes that regulate antigen presentation (*Ptprc*, *Cd86*, and MHCI genes i.e., *H2-T22*, *H2-K1*) and similar to monocytes, was significantly overrepresented for chemokine genes at day 1 (i.e., *Ccl4, Ccr5*) (Fig. 2e). Previous studies of tDCs have suggested they play a functional role in T cell antigen presentation and activation[34]. Our results, therefore, shed light on the differential roles of these innate cell subsets in the dLN early after activation.

**3M-052-Alum/OVA induces a transcriptional signature on Day 1 similar to YF-17D.** To gain a better understanding of the mechanism of action of 3M-052-Alum/OVA, we compared the transcriptomic changes after 3M-052-Alum/OVA immunization with that of the yellow fever vaccine (YF-17D). In comparison with 3M-052, which specifically activates TLR7/8, and Alum, whose mechanism of action includes NLRP3 inflammasome and STING-mediated pathways[43,44], YF-17D is known to signal through multiple innate sensors, including TLR2, TLR7, 8 and 9, RIG-I, MDA-5[27]. We isolated a total of 15,330 cells from the dLNs at baseline and day 1 post-immunization with YF-17D using the approach outlined above. Myeloid cell subsets, ILCs, NK cells, and a small CD8+ T lymphocyte population defined upon clustering and UMAP embedding corresponded to those after 3M-052-Alum/OVA immunization (Fig. 3a). Similar to the day 1 response to 3M-052-Alum/OVA, monocytes exhibited the highest number of DEGs (852 genes), followed by NK cells (Fig. 3b). This suggests that YF-17D induced a more robust NK

cell response as compared to 3M-052-Alum/OVA, in concordance with the lack of TLR7/8 expression on NK cells.

3M-052-Alum/OVA and YF-17D induced similar gene expression changes in myeloid cells (monocytes, XCR1- and XCR1+ DCs, pDCs, and tDCs) on day 1 after immunization (Pearson's R = 0.91, Fig. 3c). However, 3M-052-Alum/OVA induced increased fold changes across genes compared to YF-17D (Fig. 3d). Similar to 3M-052, gene expression profiles after YF-17D immunization were strikingly characterized by a global interferon and antiviral response, defined by TLR and IFN signaling genes, such as *Irf7*, and effector ISGs, such as *Isg15*, across nearly all myeloid and lymphoid subsets (Fig. 3e). YF-17D induced a myeloid-specific response, upregulating cell survival genes (*Serpina3g, Dnaja1*) and viral-inducible genes (*Sat1, Usp25*). As with 3M-052, YF-17D induced a lymphoid-specific response defined by cytokine-regulated genes (Fig. 3e). Previous studies have shown early activation and IFN-γ production of γδT cells in the dLN of mice as early as day 1 after YF-17D, in accordance with our findings[45]. To specifically identify unique patterns of differential gene expression between 3M-052 and YF-17D, we took DEGs (absolute logFC > 0.25, FDR < 0.05) in each group, and examined the percentage of DEGs expressed in the same direction in both groups, regardless of FDR. Although a high proportion of DEGs was shared between both 3M-052 and YF-17D groups, 3M-052 induced additional genes involved in inflammatory and IL-1β production in monocytes and XCR1- DCs (Supplementary Fig. 3).

Next, we compared the transcriptomic profile in mice with that of previously published human transcriptomic data after YF-17D vaccination[29,46]. We compared the timepoints that demonstrated

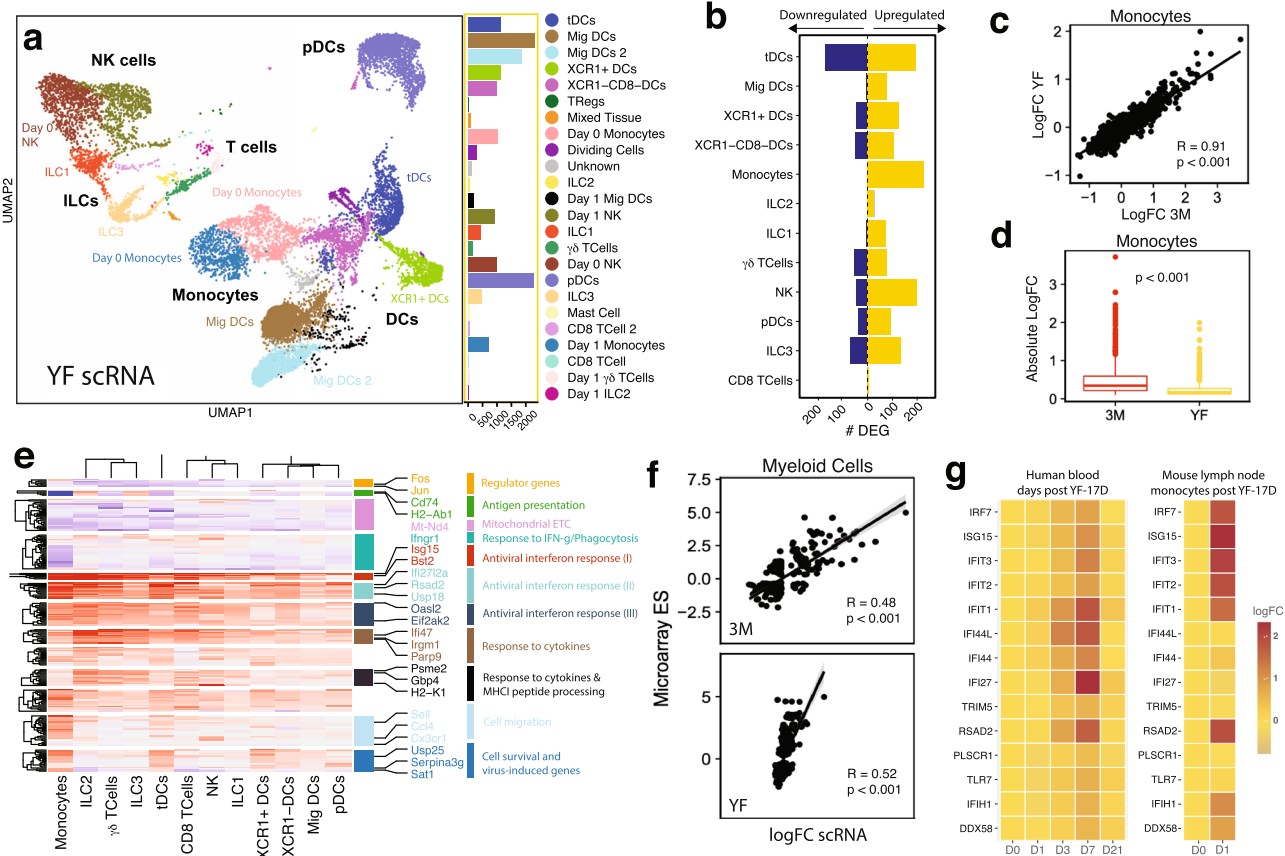

**Fig. 3 3M-052-Alum/OVA induces a transcriptional signature on Day 1 similar to YF-17D. a** UMAP embedding of 15,330 cells classified by cell type. **b** Number of upregulated (yellow) and downregulated (blue) genes in each celltype, FDR < 0.05 and absolute LogFC > 0.25. **c** Correlation of the logFC of 3M-052-Alum/OVA and YF across monocytes. $R$ = Pearson's correlation; $p$ represents two-tailed $P$ values. **d** Absolute logFC of each gene in monocytes in 3M (red) and YF (yellow) ($n = 20967$). Statistical test performed was two-sided t-test. Center line of boxplot corresponds to the median, the bounds are the 75th and 25th percentiles (interquartile range; IQR), and the whiskers are the largest or smallest value no further than 1.5*IQR from the bounds. **e** Log fold change of genes from (**b**) across all cell types. **f** Correlation between the logFC of each gene in human samples 7 days after immunization with the logFC of the corresponding homolog in mouse myeloid cells 1 day after YF-17D immunization. $R$ = Pearson's correlation; $p$ represents two-tailed $P$ values. **g** Genes previously identified as significantly changing in humans after YF-17D immunization. Left, human transcriptional changes after YF-17D vaccination; Right, corresponding changes in mouse monocytes after immunization.

the largest transcriptional changes in both mice (1 day post-vaccination) and human (7 days post-vaccination). Gene expression changes in murine dLN myeloid cells significantly correlated with bulk gene expression changes at day 7 in human PBMCs (Pearson's $R = 0.52$, $p < 0.001$) (Fig. 3f). Myeloid cells from the dLN of 3M-052-Alum/OVA treated mice also demonstrated a significant correlation with YF-17D vaccinated human PBMCs (Pearson's $R = 0.48$, $p < 0.001$), further highlighting the similarity between the two stimuli in myeloid cell activation (Fig. 3f). Consistent with previous findings in humans, we found elevated expression of interferon-stimulated genes, such as *Irf7*, *Isg15*, and *Ddx58*, after YF-17D immunization (Fig. 3g). This data demonstrates the relevance of the transcriptional changes seen in the dLN of mice to human transcriptional responses to YF-17D.

**scATAC-seq reveals differential TF motif accessibility between 3M-052-Alum/OVA and YF-17D.** To investigate the regulatory mechanisms mediating transcriptomic changes observed at day 1 in dLN innate populations, we performed scATAC-seq after 3M-052-Alum/OVA and YF-17D immunization. After filtering for low quality cells, we isolated 14,591 and 30,962 cells from the 3M-052-Alum/OVA and YF-17D immunized mice, respectively. Clustering and UMAP embedding revealed major innate subsets that were highly concordant to those identified in scRNA-seq (Fig. 4a).

We computed gene accessibility with the strategy implemented in ArchR and TF motif openness using ChromVar[47,48]. Differential analysis was then performed with correction for transcription start site (TSS) enrichment and the number of unique fragments across the cell types. Consistent with our results from scRNA-seq, monocytes exhibited the largest number of differentially accessible genes (DAGs) with 3576 DAGs (FDR < 0.05 and absolute log fold-change > 0.1) 1 day after 3M-052-Alum/OVA and YF-17D immunization, followed by tDCs with 1377 DAGs (Fig. 4b). The open myeloid cell chromatin profile significantly correlated with transcriptomic changes found in myeloid cells 1 day after immunization ($R = 0.51$, $p < 0.001$; Fig. 4c). This high epigenome and transcriptomic correlation underscores the reproducibility of our findings.

While the majority of genes were equally available and expressed, we observed a subset of genes that were more available in scATAC-seq but were not expressed to the same degree in scRNA-seq. In particular, *Ifit3* showed an increase in expression (logFC at day 1 = 2.73) and chromatin accessibility at day 1 (logFC after 24 h: 0.361) (Fig. 4c). Conversely, *Il10* gene was more accessible at day 1 (LogFC = 0.456) but was not significantly expressed (Fig. 4c). This suggests that open chromatin loci might not necessarily translate into active gene expression for some subsets of genes, but may indicate that these genes are "poised" for expression. Further

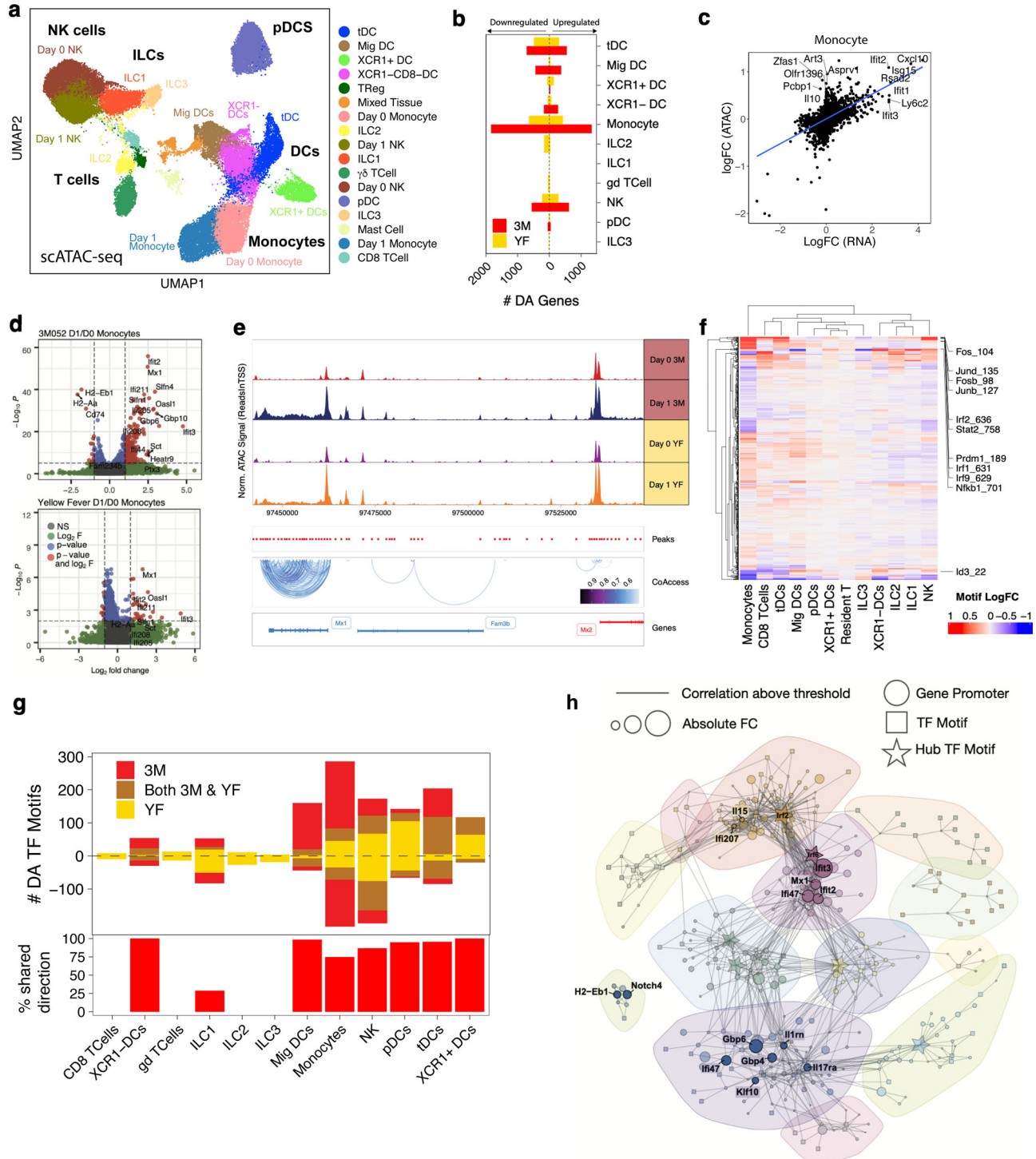

**Fig. 4 scATAC-seq reveals differential TF motif accessibility between 3M-052-Alum/OVA and YF-17D. a** UMAP embedding of 44327 cells from innate immune cells immunized with 3M-052-Alum/OVA or YF-17D. **b** Differentially accessible genes, FDR < 0.05 and absolute logFC > 0.1. **c** Correlation between logFC in monocytes in scATAC and scRNA-seq. *R* is Pearson's correlation. **d** Volcano plots of gene accessibility logFC in monocytes in 3M-052-Alum/OVA and YF-17D. **e** Representative tracks of an Irf inducible site on Chromosome 16, spanning genes Mx1 and Mx2. Peaks are shown in red below the track. **f** TF logFC 1 day after 3M-052-Alum/OVA immunization. **g** Top, differentially accessible TF motifs (FDR < 0.05) 1 day after immunization. Bottom, the percentage of 3M-052-Alum/OVA DA TF motifs that change in the same direction after YF-17D immunization. **h** The accessibility correlation network of genes and transcription factors in day 1 monocytes. Each node is a differentially accessible gene or TF; edges represent the correlation between nodes.

sampling of expression at varying timepoints could elucidate genes demonstrating different "waves" of activation.

In line with the findings in scRNA-seq, myeloid cells generated similar epigenetic responses to 3M-052-Alum/OVA and YF-17D.

For both stimuli, many of the DAGs identified on day 1 were genes involved in inflammatory and interferon responses. The most accessible DAGs in myeloid cells one day post vaccination also corresponded to most expressed DEGs from scRNA-seq.

For example, this phenomenon was true in monocytes (e.g., *Ifit2, Cxcl10,* and *Mx1*), migratory DCs (e.g., *Ifit3, Irf7,* and *Ddx60*), XCR1+ DCs (e.g., *Junb, Ifi207, Irf7,* and *Nfkbia*), XCR1- DCs (e.g., *Mx2, Stat1*), and tDCs (e.g., *Irf7/9, Ifi204,* and *Parp14*) (Fig. 4d; Supplementary Fig. 4a). In addition, the accessible chromatin regions of many ISGs, such as *Mx1*, overlapped between 3M-052-Alum/OVA and YF-17D immunized groups (Fig. 4e). This data highlights the enrichment of interferon and inflammatory genes present at both the epigenetic and transcriptomic levels after vaccination.

Next, we analyzed changes in TF motif accessibility that could reveal the regulatory pathways driving transcriptional changes found 1 day after vaccination. TF motif accessibility was computed using ChromVar. There was an increase in the accessibility of TF motifs that regulate type I interferon and pro-inflammatory pathway upon TLR activation, including Irf motifs, Fos, Jun, Nfkb1, Prdm1 across monocytes, XCR1−/+ DCs, tDCs, ILC1, ILC2, and NK cells (Fig. 4f). Overrepresentation analysis of DEGs on day 1 in monocytes and tDCs using TRRUST TF-gene database further validated the involvement of these TFs in day 1 response[49] (Supplementary Fig. 4b).

Notably, 3M-052-Alum/OVA and YF-17D shared largely similar motif profiles on day 1 across DC subsets (Pearson's R > 0.7, p < 0.001) (Supplementary Fig. 4C). This further suggests that myeloid cells exhibit common type I interferon and inflammatory signatures in the initial timepoints after 3M-052-Alum/OVA and YF-17D immunization. To dissect differences in TF activity between 3M-052-Alum/OVA and YF-17D, we calculated the number of differentially accessible TF regions (DARs) that were common or unique to each vaccination. A large proportion of DARs (>75%) across the myeloid cell subsets and NK cells were modulated in the same direction after 3M-052-Alum/OVA and YF-17D immunization (Fig. 4g). However, monocytes exhibited more unique DARs in 3M-052-Alum/OVA group, whereas more unique DARs were shown in NK cells in YF-17D group, suggesting a variable regulatory response after each immunization. A closer look into the unique differential motifs in monocytes revealed that many DARs between 3M-052-Alum/OVA and YF-17D appeared to be associated with myeloid lineage development (Supplementary Fig. 4d). For example, Cebp and Klf family motifs and Tcf4, Mesp1/2, Id3 motifs were more accessible after immunization with 3M-052-Alum/OVA and YF-17D, respectively (Supplementary Fig. 4d). These motifs have been shown to contribute to cell differentiation by controlling myeloid or lymphoid priming in lymphoid-primed multipotent progenitors[50]. Taken together, these data suggest that while 3M-052-Alum/OVA and YF-17D induced a largely comparable epigenetic profile in innate immune cells, there were unique lineage TF motifs induced in monocytes by 3M-052-Alum/OVA.

As monocytes exhibited the largest TF motif and gene accessibility changes, we further investigated the changes within monocyte subsets to reveal key TF and gene relationships. To this end, we constructed a correlation network with all DAGs and DARs, defined as features with LogFC > 0.2 and FDR < 0.005 identified on day 1. Gene-gene and gene-TF pairs with high correlations (Pearson's R > 0.9*) were connected via an edge. A community detection algorithm was then used to define modules of connected genes and TFs. Hub transcription factors were those that were connected to a disproportionally high number of genes. We found that the biggest and most connected TF-gene modules within day 1 monocytes were associated with antiviral effector genes and ISGs, and they were linked to key TF Irfs (Fig. 4h). In addition, the majority of ISGs were interconnected and could be found within a single module. Other modules constituted genes

involved in MHCII presentation, and innate immune activation, such as *Il17ra* and *Cxcr4* genes, which were linked to the Foxp1 TF motif (Fig. 4h).

## 3M-052-Alum/OVA induces long-lasting changes in innate immune cells that persisted up to Day 28.

Given the rapid response time of the innate immune system, many studies of adjuvants or vaccines in innate immune cells have primarily focused on early timepoints. Little is known about the innate immune mechanisms induced by adjuvants at later timepoints. In addition, recent studies have demonstrated that innate immunity may retain lasting epigenetic changes after vaccination that are associated with functional changes, a phenomenon referred to as trained immunity[51–54]. To investigate these phenomena in this dataset, we assessed innate immune activity on the transcriptional and epigenomic level 28 days after immunization with 3M-052-Alum/OVA and YF-17D.

First, we determined DEGs and DARs in each subset at day 28 after vaccination. To account for the possibility of a relatively attenuated immune response at day 28 compared to day 1 post-immunization, we lowered our threshold for DAGs and DEGs to those with an FDR less than 0.05 and absolute log fold change greater than or equal to 0.1. Strikingly, cell populations in 3M-052-Alum/OVA showed persistent changes even at day 28 post-immunization (Fig. 5a), particularly in the myeloid cell subsets. Monocytes had the highest number of DARs and the second most DEGs on day 28, suggesting transcriptional and epigenetic reprogramming as previously demonstrated in cells of the monocyte/macrophage lineage[32,54,55]. On the contrary, in YF-17D, there was a lack of transcriptional and epigenetic changes across myeloid cells, suggesting that the innate response had largely returned to baseline by 28 days after vaccination. Interestingly, NK cells appeared to retain epigenetic changes at day 28 after YF-17D immunization, shown by the relatively high number of DARs (Fig. 5a). As 3M-052 had been designed for slow dissemination at the local site of injection, it is possible that the persistence of the adjuvant could have contributed to the residual gene expression observed on day 28.

To better interpret the residually expressed genes 28 days after immunization with 3M-052-Alum/OVA, we used an over-representation analysis of DEGs on BTMs. Similar to day 1 post-immunization, multiple pathways involving type I interferon signaling and antiviral response were elevated at day 28 after immunization (Fig. 5b, c), implying that there was persistent activation after 3M-052-Alum/OVA immunization. Specifically, a prolonged interferon/antiviral response was detected in monocytes, multiple DC and NK populations. Importantly, increased expression of antiviral genes was accompanied by increased chromatin accessibility in the same genes (Fig. 5c). As pDCs selectively express high level of TLR7, we analyzed the differential response in LN pDCs between 3M-052 and YF-17D at both early and late timepoints after immunization. Both 3M-052 and YF-17D induced genes involved in interferon and antiviral response in pDCs, such as *Irf7* and *Ifi44*, at day 1 post-immunization. By day 28 post-immunization, some genes remained differentially expressed, notably *Irf7* and *Ifi27l2a*, after 3M-052-Alum/OVA immunization but were undetectable in LN pDCs from YF-17D group. In addition, many of the differentially expressed genes at day 28 in YF-17D were found to be ribosomal proteins, suggesting a lack of differential response at day 28 (Supplementary Fig. 5a).

Next, we sought to tease out the regulatory mechanisms of ISGs found on day 28 after 3M-052-Alum/OVA immunization. *Ifit3* was one of the ISGs that was broadly present in the cell types with sustained expression of ISGs on day 28. Thus, we correlated

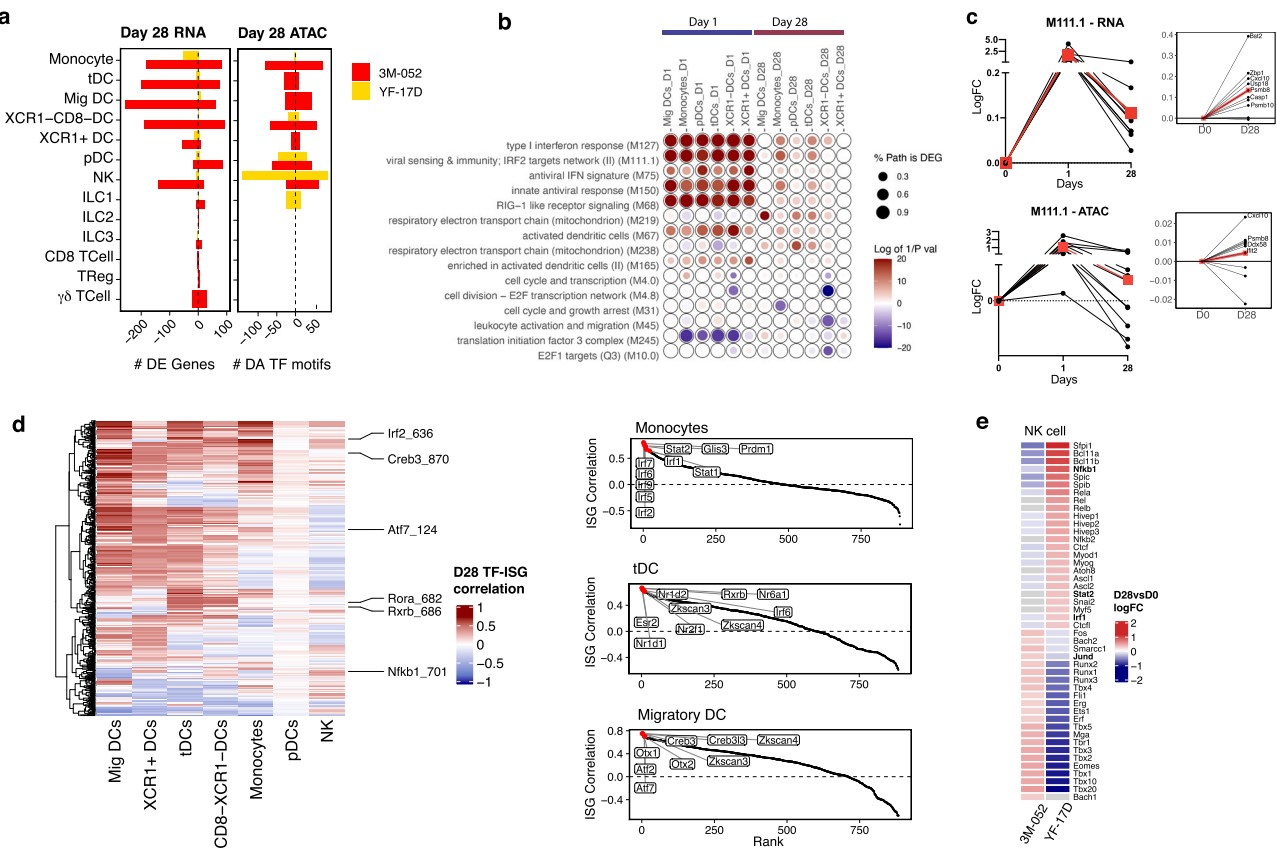

**Fig. 5 3M-052-Alum/OVA induces long-lasting changes in innate immune cells that persisted up to Day 28. a** Number of significantly expressed (RNA) genes that remain up or downregulated compared to baseline (FDR < 0.05 absolute logFC > 0.1) and the total number of significantly accessible (ATAC) motifs at day 28. **b** BTMs significantly enriched (FDR < 0.001) after 1- and 28-days post-immunization with 3M-052-Alum/OVA across dLN innate immune cells. Overrepresentation analysis of the differentially expressed genes was used to determine significance. **c** Temporal expression pattern of all genes within modules M111.1 and M111.0 (viral sensing & immunity; IRF2 targets network) in both RNA (left panels) and ATAC (right panels, red line, and square represent the mean fold change). **d** Heatmap of TF-ISG correlation in each cell subset. Pearson correlation is shown here (left). Rank plots of TF motifs that correlate with lfit3 expression in each cell type (right). **e** Top differentially accessibile TF motifs at day 28 in YF-17D compared to 3M-052-Alum/OVA (FDR < 0.05 absolute logFC > 0.1).

TF motif z-score with *Ifit3* gene accessibility score within single cells at day 28 (Fig. 5d). Strikingly, we found that Irf2 and Stat2 motif openness strongly correlated with *Ifit3* gene accessibility in monocytes (Irf2, Pearson's $R = 0.803$; Stat2, $R = 0.725$). Irf (Interferon response factor) and Stat (signal transducer and activator of transcription) transcription factors are key regulators of antiviral immunity[56].

Notably, in contrast to monocytes, Ifit3 correlation in tDCs and migratory DCs was dominated by non-Irf and Stat motifs. In particular Rxrb motif in tDCs ($R = 0.660$; In monocyte: $R = 0.040$) and Creb3 motif in migratory DCs ($R = 0.748$; In monocyte: $R = 0.435$). interestingly, Rxrb is a nuclear retinoic acid receptor involved in metabolic signaling and was previously found to play a role in type I interferon and inflammatory responses and Creb3 is an endoplasmic reticulum transcription factor associated with the unfolded protein response (UPR)[57–61]. Together, our data highlights the role of Irf- and Stat-related epigenomic reprogramming in monocytes and suggests that distinct gene regulatory mechanisms could potentially be dominant in DC subsets.

Although there was a lack of transcriptional changes in dLN innate immune cells 28 days after YF-17D immunization (Supplementary Fig. 5b), we did observe increased accessibility of TF motifs and gene chromatin loci related to interferon and inflammatory pathways in NK cells. In contrast, similar epigenetic changes that are associated with innate activation were largely undetectable in NK cells on day 28 after 3M-052-Alum/OVA

immunization. Specifically, TF motifs including Nfkb (logFC = 0.840), Irf1 (logFC = 0.311) and Bcl11a (logFC = 0.962) respectively, and chromatin loci of genes such as *Jund* (FDR-adjusted $P < 0.001$; logFC = 0.102) and *Socs1* (FDR-adjusted $P < 0.001$; logFC = 0.096) (Fig. 5e; Supplementary Fig. 5c). This implies epigenetic imprinting of NK cell activation, which has been previously shown in mouse memory NK cells after viral infection, and suggests that epigenetic memory in NK cells could potentially be induced by a live vaccine[62].

**Differentially expressed and accessible genes in 3M-052-Alum/OVA are driven by different sub-clusters of monocytes on Day 28.** To determine the transcriptional and epigenomic changes observed at a single cell level, especially in monocytes, 28 days post-immunization with 3M-052-Alum/OVA, we sub-clustered and re-embedded the monocytes profiled at baseline, day 1 and 28 (Fig. 6a, b). Based on transcriptomics data, monocytes separated into distinct subclusters based on vaccination time, with day 1 monocytes separated out from day 0 and 28 monocytes (Fig. 6a). Day 1 monocytes consisted of a heterogenous pool of activated Ly6Chi monocytes, defined by distinct sets of immune activation and antiviral genes, including *Isg15*, *Ccl5*, *Cfb*, and *Anxa1* and *Ccl2* (Supplementary Fig. 6a). A further sub-clustering of day 0 and 28 monocytes revealed 3 sub-clusters that appeared to represent major subsets of murine blood monocytes corresponding to their gene signatures. Classical (Ly6C$^{hi}$) monocyte

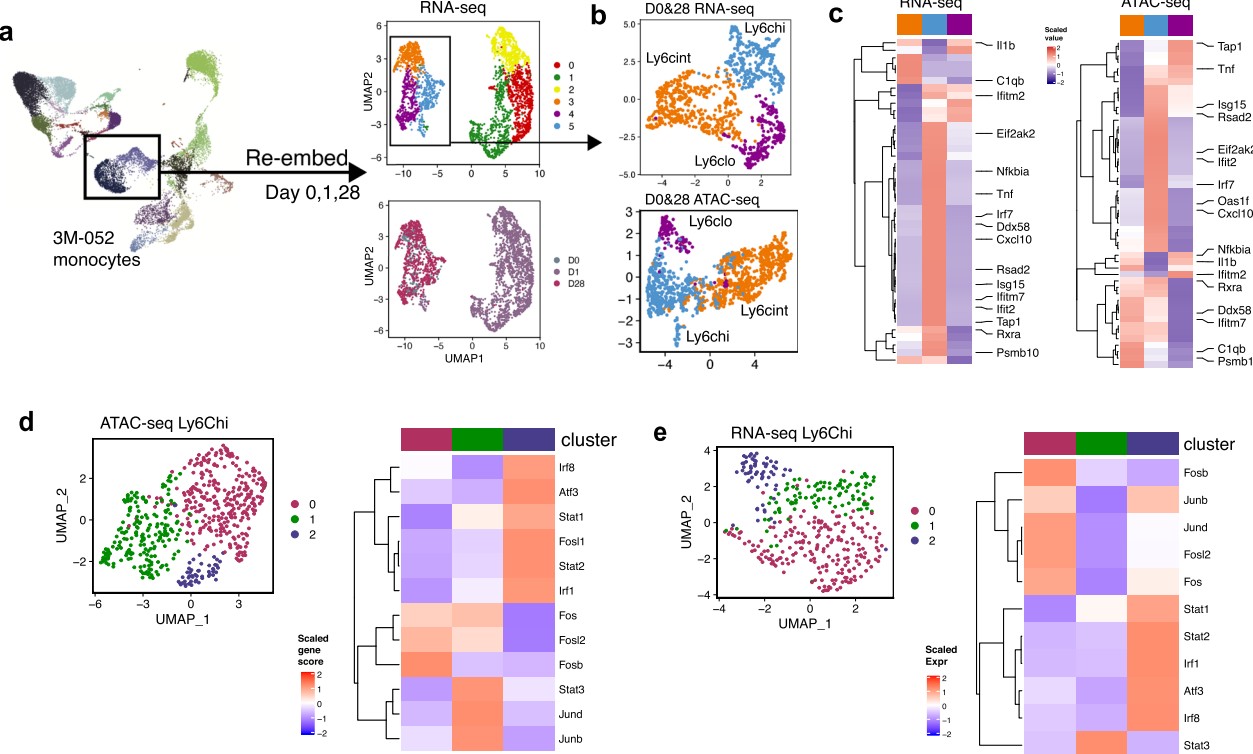

**Fig. 6 Differentially expressed and accessible genes in 3M-052-Alum/OVA are driven by different sub-clusters of monocytes on Day 28. a** Re-embedding of monocytes present before (day 0) and 1 and 28 days after immunization with 3M-052-Alum/OVA in scRNA-seq. **b** Sub-clustering and re-embedding of monocytes at baseline and 28 days after 3M-052-Alum/OVA immunization in scRNA-seq (top) and scATAC-seq (bottom). **c** Gene expression and accessibility score of antiviral BTM module highlighted in Fig. 5b, across day 0 and day 28 monocyte subclusters. **d** Re-clustering of Ly6Chi monocyte subcluster identified in (**c**) (left); heatmap of Irf and AP-1 gene accessibility within Ly6Chi monocyte subclusters (right), performed on scATAC-seq data. **e** Re-clustering of Ly6Chi monocyte subcluster identified in (**c**) (left); heatmap of Irf and AP-1 gene accessibility within Ly6Chi monocyte subclusters (right), performed on scRNA-seq data.

subcluster was defined by expression of *Ly6c2*, *Ccr2*, whereas intermediate (Ly6C$^{int}$) and non-classical (Ly6C$^{lo}$) monocytes expressed MHCII-related genes like *Cd74*, *H2-Aa*, *Ciita* and *Cx3cr1*, *Nr4a1* and *Pparg* respectively, based on previous reports[41,63,64] (Fig. 6b; Supplementary Fig. 6b). On the epigenomic level, sub-clustering analysis of monocytes at day 0 and 28 after immunization reproduced similar Ly6C$^{hi}$, Ly6C$^{int}$, Ly6C$^{lo}$ clusters that were defined by a near identical set of unique genes (Fig. 6b; Supplementary Fig. 6b).

Most recently, another study from our lab looking at a squalene-based adjuvant, AS03, in the context of influenza vaccination in human, found a pattern of elevated Irf and reduced AP-1 motif accessibility in classical monocytes[32]. Notably, Irf motifs were amongst the highest correlated TF motif with ISG activation in monocytes in our data set (Fig. 5d). We first asked whether the elevated accessibility and expression of ISGs observed on day 28 could be attributed to distinct subclusters of monocytes. Interestingly, analysis of the transcriptomic and epigenome of these cells revealed that amongst the subclusters, Ly6C$^{hi}$ monocytes displayed the highest expression and accessibility of antiviral and interferon-stimulated genes (Fig. 6c). Based on cellular kinetics analyzed by flow cytometry, Ly6C$^{hi}$ monocytes increased in proportion amongst the total monocytes identified on day 28 post-immunization (Supplementary Fig. 6c). In addition, both Ly6C$^{int}$ and Ly6C$^{hi}$ subclusters displayed enhanced antiviral gene program on a single cell level (Supplementary Fig. 6c). Taken together, this suggests that the sustained expression of ISGs on day 28 could be attributed to: (1) a higher proportion of Ly6Chi cells, and (2) activation in Ly6Cint cells on a single cell level.

As epigenetic reprogramming was previously linked to a subset of CD14+ CD16− classical monocytes[32], which are equivalent of murine Ly6C$^{hi}$ monocytes, we questioned if this distinct subcluster could also be observed in the mouse system. To this end, we performed sub-clustering of Ly6C$^{hi}$ monocytes identified in our scATAC-seq dataset. This approach separated the Ly6C$^{hi}$ cells into 3 distinct clusters that exhibited differential proportion (Fig. 6e). Notably, cluster 2 exhibited a similar pattern of gene accessibility and resembled the classical monocyte subpopulation identified at a late immunization time point in human PBMCs[32]. In addition, this subcluster was further reiterated in transcriptomic analysis (Fig. 6d). These findings illustrate the heterogeneity of monocytes in murine lymph nodes, delineated based on transcriptomics and epigenome, following vaccination, and highlights distinct subclusters in retaining a primed antiviral state at a late timepoint following vaccination.

## Discussion

Novel adjuvants hold the potential to dramatically increase the efficacy of widely used vaccines, such as those against influenza or COVID-19. A newly developed class of adjuvants that is thought to hold great promise target PAMPs, strongly and specifically activating the innate immune system. Here, we use scRNA-seq and scATAC-seq to delineate the single cell epigenetic and transcriptomic signatures induced by a novel TLR7/8 agonist, 3M-052, in the dLN of mice. We additionally measured the response to YF-17D, a model vaccine that confers long-lasting protection with a single dose. The cellular kinetics, transcriptomic and epigenetic data one day after immunization indicate that

both 3M-052-Alum/OVA and YF-17D elicit a qualitatively similar response across a broad range of innate immune cells. The immune response induced by both YF-17D and the adjuvant is primarily defined by the upregulation of a repertoire of type I IFN and antiviral genes. Further investigation of chromatin accessibility after immunization revealed increased openness of transcription factor motifs, such as Irf1/2, that are known to be key players in the TLR signaling and type I IFN pathway. The shared molecular signature between 3M-052-Alum/OVA and YF-17D suggests that both vaccines might share an overlapping molecular mechanism of action during the initial hours post-immunization. Additionally, 3M-052-Alum/OVA appeared to induce more unique differentially expressed genes and accessible motifs across monocytes and specific DC subsets compared to YF-17D, suggesting that it acts primarily through these myeloid cells.

Myeloid cells continued to display transcriptomic and epigenomic activation 28 days after immunization with 3M-052-Alum/OVA. This activation could be the result of persistent activation from initial immunization, or could be due to continuous innate activation by the slow releasing adjuvant. In either case, this finding highlights the potency of 3M-052-Alum/OVA. The residual activation also provides intriguing evidence to support the notion of vaccine-induced trained immunity. An important follow-up to this finding would be to further investigate whether the residual response has implications for subsequent vaccination or infection. On the contrary, we did not detect any persistence of activation in myeloid cells and NK cells on the transcriptomic level following YF-17D immunization on day 28. However, the epigenomic analysis revealed that 28 days after YF-17D immunization, NK cells retained elevated accessibility of Nfkb and Irf motifs, implying that these cells might be epigenetically poised for subsequent activation. The persistent activation of myeloid cells and NK cells at day 28 post-immunization in 3M-052-Alum/OVA and YF-17D, respectively, also highlights an aspect of differential immune response elicited by the two stimuli.

The integration of scRNA-seq and scATAC-seq also enabled us to delineate the major transcription factors that may drive the residual activation of ISGs on day 28 after 3M-052-Alum/OVA immunization. We uncovered putative TF-gene targets within distinct cell subsets. Namely, nuclear retinoic acid receptor motif, Rxrb, which had been implicated in innate interferon signaling or inflammatory pathways, in DC subsets, as well as Irf2 and Stat2 motifs in monocytes. It will therefore be of interest to validate these signaling pathways in specific genetic knockout experiments.

Of note, our data sets revealed the heterogeneity of monocytes in murine dLNs on a transcriptomic and epigenomic level. In addition, the activation in monocytes seen at a later stage of immunization (28 days) with 3M-052-Alum/OVA was contributed by distinct monocyte subsets. Transcriptomic analysis revealed that Ly6Chi monocytes had a high baseline expression of various ISGs and increased in frequency, whereas Ly6Cint and Ly6Clo monocytes retained ISG expression at day 28. In particular, a distinct subcluster of classical monocytes with a pattern of enhanced Irf and reduced AP-1 gene accessibility and expression that was identified resembles a monocyte subcluster previously reported in human PBMCs after the administration of an adjuvanted vaccine[32]. This further suggests that monocytes could contribute to epigenetic reprogramming at a late stage of vaccination[51,52,54], specifically one that is induced in part by an adjuvant.

Lastly, in addition to profiling the strong response of myeloid cells after immunization, we found that lymphoid cells in the dLN, in particular ILCs, NK cells, and gamma-delta or CD8+ T cells, also exhibited an elevated antiviral type I IFN response in the initial timepoint post-immunization. This suggests that they play an important functional role in priming the ensuing immune response as they are known to contribute to early cytokine production in tissue microenvironment[65]. Altogether, our scRNA-seq and scATAC-seq data provide a comprehensive map of the early and late immune response during vaccination and can serve as a rich data set for future vaccine studies.

## Methods

**Mice and immunization.** C57BL/6 mice were purchased from Jackson Laboratories. All mice used were female mice aged between 8 and 14 weeks. For immunization, mice were injected subcutaneously at the base of the tail with 100 μl of 3M-052-Alum adjuvant mixed with 25 μg OVA or with $10^6$ PFU YF-17D. All mice in this study were maintained under specific-pathogen-free conditions in the Stanford Research Animal Facility. All animal studies were conducted by following animal protocols reviewed and approved by the Institutional Animal Care and Use Committee of Stanford University.

**YF-17D virus and 3M-052.** The YF-17D virus was a gift from R. Ahmed (Emory University, Atlanta, GA). YF-17D was cultured in Vero cells in DMEM supplemented with 10% FBS and penicillin/streptomycin. After ~6 days post-infection, cell culture supernatants were collected and frozen at 80 °C in aliquots. Viral titer was quantitated by plaque assay using Vero cells. 3M-052-Alum was obtained from 3M company.

**Lymph node processing and flow cytometry.** Inguinal lymph nodes were harvested and treated with 5 mg/ml collagenase type IV (Worthington) for 20 min at 37 °C, followed by smashing with a 100 μm strainer to obtain single cell suspension. Samples were then stained with Zombie UV™ (1:200; BUV496; Biolegend #423107), anti-Ly6C (1:500; BV780; Biolegend #128041), anti-Ly6G (1:400; APC-Cy7; Biolegend #127624), anti-CD19 (1:100; BB700; BD #566411), anti-CD3 (1:100; BB700; BD #742175), anti-MHCII (1:400; AF700; eBioscience #56-5321-82), anti-CD11b (1:300; BV650; Biolegend #101239), anti-CD11c (1:400; BV421; Biolegend #117330), anti-CD86 (1:300; A647; Biolegend #105020), anti-Siglec-F (1:400; PE-CF594; BD #562757), anti-CD24 (1:200; BUV395; BD #744471), anti-CD45 (1:200; BV610; Biolegend #103140), anti-CD169 (1:200; PE-Cy7; Biolegend #142412), anti-PDCA-1 (1:200; BUV563; BD #749275), anti-CD8a (1:200; BUV805; BD #612898), anti-CD103 (1:100; PE; eBioscience #12-1031-82), anti-NK1.1 (1:200; BV510; Biolegend #108738), anti-F4/80 (1:100; BUV737; BD #749283), anti-CD205 (1:100; eBioscience #53-2051-82), Ghost Dye™ Violet 510 (1:400; Tonbo Bioscience #13-0870-T100), anti-CD95 (1:200; PE-Cy7, clone:Jo2, BD Biosciences #557653), anti-CD19 (1:200; PerCP-Cy5.5, clone:1D3/CD19, BioLegend #152406), anti-CD38 (1:200; BUV395, clone:90, BioLegend #102702), anti-CD4 (1:100; BV650, clone:GK1.5, BioLegend #100469), anti-CXCR5 (1:50; BV711, clone:L138D7, BioLegend #145529), anti-PD1 (1:200; PE-Dazzle594, clone:29F.1A12, BioLegend #135228). Samples were then washed twice with PBS + 2% FBS + 1 mM EDTA and fixed with BD Cytofix (#554655). Samples were analyzed on a BD FACS Symphony analyzer with BD FACS Diva v.8.0.1. In data with comparison to a control group across multiple timepoints, two-way ANOVA with Dunnett's multiple comparison test was performed. Statistical tests and plotting of flow cytometry data were performed using GraphPad Prism 9.

**ELISA.** Following vaccination, mice were bled at indicated time points for serological analysis. Serum was obtained by centrifuging whole blood collection in micro sample tube with serum gel (SARSTEDT). ELISA plates were coated overnight at 4 °C with 100 μL of 10 μg OVA for detecting OVA-specific antibodies. StartingBlock™ (PBS-T) blocking buffer (Thermo Scientific) was used for blocking and sample dilution. Specific isotypes were identified using HRP-conjugated isotype-specific anti-mouse antibodies obtained from SouthernBiotech, and developed using Pierce TMB substrate kit. ELISA data were analyzed using an endpoint analysis. Serial dilution of pooled positive control samples was used to produce an assay sensitivity curve (Prism: Asymmetric Sigmoidal, 5PL, X is log), and biological samples were compared to that curve to assign a titer (AU) relative to the assay's lower threshold (calculated as the average value of six blank wells plus 3 times their standard deviation). Groups were then compared by standard statistical testing using Prism statistical analysis software.

**Magnetic isolation and FACS sorting of LN cells.** Inguinal lymph nodes were harvested and processed to obtain single cell suspension. Total LN cells were then stained with anti-CD16/32 antibodies for blocking of Fc receptors, followed by staining with biotinylated anti-CD3 (Biolegend #100244) and anti-CD19 (eBioscience #13-0193-82) antibodies for 20 min at 4 °C. Next, samples were incubated with streptavidin-conjugated magnetic beads (BD #557812) for 30 min at 4 °C. and passed through a magnet (STEMCELL technologies), according to the manufacturer's protocol. The negative fraction was transferred into a clean tube and stained for FACS sorting. Antibodies used for sorting include Live/Dead Fixable Aqua (BV510; Tonbo Biosciences #13-0870-T100), anti-CD19 (APC; Biolegend #152410), anti-CD3 (APC; Biolegend #100236), anti-CD45 (BV610;

Biolegend #103140), anti-CD11c (BV421; Biolegend #117330), anti-CD11b (FITC; Biolegend #101206), anti-Ly6C (BV780; Biolegend #128041), anti-Ly6G (APC-Cy7; Biolegend #127624), anti-Siglec-F (PE-CF594; BD #562757), anti-PDCA-1 (PE; Biolegend 127010). Samples were sorted on a FACSAria Fusion instrument.

**scATAC-seq.** FACS-purified cells were processed for single nuclei ATAC-seq according to the manufacturer's instructions (10x Genomics, CG000168 Rev D). Briefly, nuclei were obtained by incubating cells for 4 min in freshly prepared Lysis buffer following manufacturer's instructions for Low Cell Input Nuclei Isolation (10x Genomics, CG000169 Rev C). Nuclei were washed and resuspended in chilled diluted nuclei buffer (10x Genomics, 2000153). Next, nuclei were subjected to transposition for 1 h at 37 °C on the C1000 touch PCR instrument (BioRad) prior to single nucleus capture on the 10x Chromium instrument. Samples were subjected to post GEM cleanup, sample index PCR, cleanup, and library QC prior to sequencing. Samples were pooled, quantified, and sequenced on the HiSeq 4000 instrument (Illumina).

**scRNA-seq.** FACS-sorted cells were resuspended in cold PBS supplemented with 1% BSA (Miltenyi) and 0.5 U/μL RNase Inhibitor (Sigma Aldrich). Cells were partitioned into Gel Beads-in-emulsion (GEMs) using the 10x Chromium 3' V3 chemistry system (10x Genomics). The released RNA was reverse transcribed in the C1000 touch PCR instrument (BioRad). Barcoded cDNA was extracted from the GEMs by post-GEM RT-cleanup and amplified for 12 cycles. Amplified cDNA was subjected to 0.6x SPRI beads cleanup (Beckman, B23318). 25% of the amplified cDNA was subjected to enzymatic fragmentation, end-repair, A tailing, adapter ligation, and 10X specific sample indexing as per manufacturer's protocol. Sequencing libraries were generated and the quality was assessed through Bioanalyzer (Agilent) analysis. Libraries were pooled and sequenced on the HiSeq 4000 instrument (Illumina) with a targeted read depth of 40,000 read pairs/cell.

**scRNA-seq analysis.** Raw count data was filtered to remove cells with a mitochondrial RNA fraction greater than 25% of total RNA counts per cell. The resultant count matrix was used to create a Seurat (v 3.1.4) object. Filtered read counts were scaled by a factor of 10,000 and log transformed. Cells with greater than 20% mitochondrial RNA were removed. The top 2000 variable RNA features were used to perform PCA on the log-transformed counts. Using a scree plot, we chose the first 25 principal components (PCs) to perform further downstream analyses, including clustering and UMAP projections. Clusters were identified with Seurat SNN graph construction followed by Louvain community detection on the resultant graph with a resolution of 0.4, yielding 24 clusters. The R package scds was used to identify doublets; we removed cells with a hybrid doublet score at or above the top 95% percentile of all cells. No batch effect was seen between yellow fever or 3M-052 treated cells at baseline[66]. Differentially expressed genes (DEGs) were identified using Seurat's FindMarkers function. Genes with an FDR < 0.05 and absolute log fold change >0.25 (after 1 day post-immunization) or 0.1 (28 days post-immunization) compared to baseline were considered significant. Overrepresentation analysis was used to identify significantly enriched BTMs and TRRUST TF within the DEGs[67]. P-values for the overrepresentation analysis were determined by hypergeometric distribution. Significant pathways were those with an FDR < 0.05. ComplexHeatmap (v.2.1.0) was used to produce all heatmaps.

**Microarray analysis of human YF-17D data.** Microarrays (GSE136163, GSE124533, and GSE13699) were downloaded from NCBI's GEO using the MetaIntegrator package[68]. We ensured all downloaded gene expression data were log2-transformed. For each gene, we calculated change in expression between pre- and post-vaccination as Hedges' g. We used the random-effects inverse variance meta-analysis using Dersimonian–Laird method to calculate a summary effect size (ES) across datasets for each gene. The effect sizes for each gene were then compared to the fold change at 1 day post-immunization in the corresponding mouse homolog.

**scATAC-seq analysis.** Raw sequencing data were converted to fastq format using cellranger atac mkfastq (10x Genomics, v.1.0.0). scATAC-seq reads were aligned to the mm10 reference genome and quantified using cellranger count (10x Genomics, v.1.0.0). All scATAC data was processed with ArchR. To remove poor quality cells, we filtered cells with less than 1000 unique fragments and enrichment at TSSs below 8. ArchR's doublet detector was run to identify clusters with a doublet score in the top 10% of all cells. One such cluster was identified and removed from downstream analysis. DAGs were identified using ArchR's getMarkerFeatures function, which selects groups of cells with comparable technical biases. ChromVAR[48], implemented from within ArchR, was used to determine the transcription factor motif availability. We used the mm10 Catalog of Inferred Sequence Binding Preferences (CIS-BP) to determine motif locations. All scores were GC bias corrected.

**Network analysis with scATAC-seq data.** To develop the gene-TF network, we analyzed the correlation between all genes and TF motifs that were differentially expressed after 24 h post vaccination. We determined gene-gene edges as those with a correlation above 0.9. Gene-TF correlations above 0.8 were also assigned an edge. The R package Igraph was used to build a graph from the resulting adjacency matrix. The size of the node (gene or TF) was determined by logFC in monocytes at 1 day post-immunization. The layout of the graph was determined with the Fruchterman–Reingold algorithm, as implemented by Igraph. Community structure was determined by fast greedy clustering. Communities with fewer than 4 nodes were removed.

**Statistical analysis.** The difference in mean between test and control groups were analyzed by two-way ANOVA test with Dunnett's multiple comparisons test for more than one timepoint (Fig. 1b, c) and one-way ANOVA test with Dunn's multiple comparisons test for one timepoint (Supplementary Fig. 1a). Two-way ANOVA with Tukey's multiple comparison test was performed for data with multiple groups and timepoints (Supplementary Fig. 1b). The difference between any two groups for flow cytometry analysis was measured using two-tailed Mann–Whitney test (Supplementary Fig. 6c). The difference between groups for scRNA-seq and scATAC-seq analysis was measured using Student t-test, with Benjamini–Hochberg correction when multiple comparisons were performed. All correlations are Pearson's correlation. When boxplots are included, center line of boxplot corresponds to the median, the bounds are the 75th and 25th percentiles (interquartile range; IQR), and the whiskers are the largest or smallest value no further than 1.5*IQR from the bounds. All statistical analyses with flow cytometry data were performed on GraphPad Prism v9. All scRNA-seq and scATAC-seq analyses were performed with Seurat (v 3.1.4) and ArchR packages in R, respectively.

**Reporting summary.** Further information on research design is available in the Nature Research Reporting Summary linked to this article.

## Data availability
scRNA-seq and scATAC-seq data are publicly accessible in the GEO under accession numbers GSE180384 and GSE180752, respectively. Source data for Fig. 1b, c are provided with this paper. Any other relevant data are available from the corresponding authors upon reasonable request. Source data are provided with this paper.

## Code availability
The codes used in the study are available in GitHub https://github.com/scottmk777/3M052Vaccine. Some codes used for meta-analysis can be obtained from the corresponding authors upon reasonable request.

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

## Acknowledgements

This study was funded by National Institutes of Health (R37 DK057665; R01 AI048638; U19 AI057266; U19 AI090023), Bill and Melinda Gates Foundation, Open Philanthropy, and the Violetta L. Horton and Soffer Endowments to B.P. P.K. is funded in part by the Bill and Melinda Gates Foundation (OPP1113682); the National Institute of Allergy and Infectious Diseases (NIAID) grants 1U19AI109662, U19AI057229, and 5R01AI125197. We acknowledge the Stanford Functional Genomics Facility for technical assistance; Dhananjay Wagh and Ed Kim for library preparation, John Coller for data analysis. Cell sorting and flow cytometry analysis were performed on instruments purchased by Parker Institute for Cancer Immunotherapy in the Stanford Shared FACS Facility. Figure 1a was created with BioRender.com.

## Author contributions

B.P. conceptualized the study; B.P. and A.L. designed the study. A.L., F.W. and P.S.A. performed scRNA-seq and scATAC-seq; A.L. and M.K.D.S. contributed equally; F.W. and P.S.A. contributed equally; A.L. performed flow cytometry of innate cells; M.K.D.S. and A.L. analyzed scRNA-seq and scATAC-seq; A.L. and M.K.D.S. performed data visualization; W.L. performed flow cytometry and analysis of B and GC, Tfh cells; M.T. and C.B.F. provided 3M-052-Alum; B.P. and P.K. supervised the project; A.L. and M.K.D.S. wrote the paper. B.P. acquired funding. All authors read and accepted the manuscript.

## Competing interests

B.P. serves on the External Immunology Board of GlaxoSmithKline, and on the Scientific Advisory Board of Medicago. M.T. is an employee of 3M, the manufacturer of 3M-052-Alum used in this study. C.B.F. is an inventor on a patent application regarding the 3M-052-Alum formulation. The remaining authors declare no competing interests.
