## [Peer Review File · Nature Communications]

A Molecular Atlas of Innate Immunity to Adjuvanted and Live Attenuated Vaccines, in MiceREVIEWER COMMENTS

Reviewer #1 (Remarks to the Author):

In this exciting work by Lee and colleagues, they address an important, emerging concept in contemporary vaccinology: namely – trained innate immunity. In particular, they address it by studying the early and mid-term impact of adjuvantation on innate immune cells and how they may contribute to the establishment of long-lasting adaptive responses. The authors use the mouse model, and vaccinate animals against OVA using a highly relevant novel adjuvant, 3M-052, that has been demonstrated to remarkably increase antibody titers and improve the durability of the antibody response for protein immunogens. This adjuvant is not licensed but is currently undergoing clinical trials for safety and efficacy. The study uses cutting edge methodology, single-cell ATAC-Seq and sc-RNA-Seq to characterize very early (Day 1) and mid-term (Day 28) changes to the genome and epigenome in distinct innate cell types. To understand signals that may be important, they perform a parallel set of experiments using the YF-17D vaccine against yellow fever virus, as it induces humoral and CD8+ T cell responses that persist for decades. Lastly – findings in recent years have indicated that immunological information obtained from cells in LN's may be much more informative than in blood, and the emphasis on LN innate cells adds additional impact. The authors find, interestingly, that the epigenetic changes induced by both 3M052 and YF-17D regimens are maintained as long as 28 days post-vaccination – despite blood-based estimates of turnover of many of these cells being much shorter. The authors also provide, in detail, the pathways, genes and operon chromatin regions in common between both vaccines. Overall, the findings are important, highly novel and likely to be of broad interest to basic immunologists and those in the translational vaccinology field. The findings take on extra importance as the field tries to understand how prior coronavirus exposure does not lead to long-term antibody responses. The methodology and bioinformatics analyses are sound, and the findings statistically significant. While there were some deficiencies (outlined below), the majority of these can be addressed by additional bioinformatic analyses and limited new experimentation. The conclusions are justified by the results.

Major points:

1. As the authors point out, the 3M052 provides part of the innate signaling via TLR8, however there are known differences in the expression of TLR8 in dendritic cells between mice and primates. The authors should provide a more detailed explanation of this limitation in the Discussion.
2. Recent data has demonstrated the gene signatures in pDCs in lymph nodes are much different than those in blood. While the authors examined pDCs, this was somewhat underdescribed. Did the pDCs express IFNA? Also, pDCs should have high levels of IRF7 but it was hard to tell if this was used as a defining marker. Overall, this reviewer would find the manuscript improved by a deeper dive on LN pDCs. Additionally, a detailed analysis of cells making IFNA vs IFNB in the LN would help improve the manuscript.
3. While there is an emphasis on pathways in common between the vaccine regimens, one area not explored is differences that may underlie the different “phenotypes” of these vaccines. In particular, YF-17D induces exceptionally long-lived cellular responses, and most of the demonstrated enhancement of 3M052 has been on the humoral responses. A more in-depth comparison of differences would be useful to generate novel hypotheses would improve the manuscript.

Minor Points:

1. There are several figures where the images are too small, and the font for Greek symbols has not rendered correctly.
2. The colour code in the UMAPs is hard to follow and could be improved by denoting cells by text in-figure.

Reviewer #2 (Remarks to the Author):

Lee, Scott and colleagues apply single-cell genomics techniques to explore the immune response in murine lymph nodes following vaccination. The study is incredibly well executed and the manuscript reads well. If there is anything that I could suggest the authors do a little more of is speculate about the mechanisms that drive the prolonged myeloid reprogramming signature at 28 days; the authors do a little bit of this but I'm curious if there could be any evidence of myeloid reprogramming in the bone marrow as others have speculated.

I recommend swift acceptance.

REVIEWER COMMENTS

Reviewer #1 (Remarks to the Author):

In this exciting work by Lee and colleagues, they address an important, emerging concept in contemporary vaccinology: namely – trained innate immunity. In particular, they address it by studying the early and mid-term impact of adjuvantation on innate immune cells and how they may contribute to the establishment of long-lasting adaptive responses. The authors use the mouse model, and vaccinate animals against OVA using a highly relevant novel adjuvant, 3M-052, that has been demonstrated to remarkably increase antibody titers and improve the durability of the antibody response for protein immunogens. This adjuvant is not licensed but is currently undergoing clinical trials for safety and efficacy. The study uses cutting edge methodology, single-cell ATAC-Seq and sc-RNA-Seq to characterize very early (Day 1) and mid-term (Day 28) changes to the genome and epigenome in distinct innate cell types. To understand signals that may be important, they perform a parallel set of experiments using the YF-17D vaccine against yellow fever virus, as it induces humoral and CD8⁺ T cell responses that persist for decades. Lastly – findings in recent years have indicated that immunological information obtained from cells in LN's may be much more informative than in blood, and the emphasis on LN innate cells adds additional impact. The authors find, interestingly, that the epigenetic changes induced by both 3M052 and YF-17D regimens are maintained as long as 28 days post-vaccination – despite blood-based estimates of turnover of many of these cells being much shorter. The authors also provide, in detail, the pathways, genes and open chromatin regions in common between both vaccines. Overall, the findings are important, highly novel and likely to be of broad interest to basic immunologists and those in the translational vaccinology field. The findings take on extra importance as the field tries to understand how prior coronavirus exposure does not lead to long-term antibody responses. The methodology and bioinformatics analyses are sound, and the findings statistically significant. While there were some deficiencies (outlined below), the majority of these can be addressed by additional bioinformatic analyses and limited new experimentation. The conclusions are justified by the results.

Major points:

1. As the authors point out, the 3M052 provides part of the innate signaling via TLR8, however there are known differences in the expression of TLR8 in dendritic cells between mice and primates. The authors should provide a more detailed explanation of this limitation in the Discussion.

We agree with the reviewer that there are differences in TLR8 expression between mice and primates and will include this in the Discussion. Of note, a previous study demonstrated that 3M-052 vaccination in non-human primates activated similar monocyte subsets in peripheral blood, denoted by CD86 expression at day 1 post-vaccination¹. This suggests parallel modes of actions 3M-052 in mice and primates, but we cannot rule out the possibility that there could be other differences in primates with a functional TLR8.

2. Recent data has demonstrated the gene signatures in pDCs in lymph nodes are much different than those in blood. While the authors examined pDCs, this was somewhat underdescribed. Did the pDCs express IFNA? Also, pDCs should have high levels of IRF7 but it was hard to tell if this was used as a defining marker. Overall, this reviewer would find the manuscript improved by a deeper dive on LN pDCs. Additionally, a detailed analysis of cells making IFNA vs IFNB in the LN would help improve the manuscript.

We had always been curious to see IFNA expression in pDCs, and it is surprising that we did not detect any IFNA or IFNB expression in our scRNA-seq data set at the day 1 and 28 timepoints sampled. However, the lack of IFN-alpha expression on a transcriptomic level is also consistent with studies profiling human and non-human primate vaccine responses at 24 and 48 hours after immunization^{2,3}.

There are a few explanations for why this could be so. One of which could be the transient expression of IFN-alpha upon stimulation (**Reviewer Figure 1**). Other studies with 3M-052 have shown that IFN-alpha was detectable in the serum and dLN of mice at 12 hours post-immunization, but not at earlier timepoint⁴. However, we did observe the activation of pDCs at day 1 post-immunization, based on their CD86 expression by flow cytometry and their expression of IFN-inducible antiviral genes (Figure 1B and 2E in manuscript).

Figure 1. Expression of type I IFN.

We observed a relatively high expression of *Irf7* in pDCs (**Reviewer Figure 2**). However, *Irf7* is also constitutively expressed in other cell types, such as monocytes⁵, and was not the key defining marker of pDCs in our analysis. When compared against other cell clusters, pDCs were most specific for expression of *Irf8*, *Siglech*, *Ly6d*. These markers have been previously used to identify pDCs⁶. Additionally, we flow sorted pDCs before sequencing based on their expression as CD3⁻CD19⁺PDCA1⁺CD11b⁻. Thus, we did not solely rely on transcriptomic markers but instead had corroborating canonical surface expression markers to accurately identify pDCs.

Figure 2. Expression of *Irf7* across cell types.

We agree with the reviewer on the deeper analysis of LN pDCs and performed further sub-clustering of pDCs in 3M-052. We found that in addition to its activation at day 1 and residual activation at day 28 post-immunization (Figure 2D and 5B in manuscript), LN pDCs remain a largely homogenous population (**Reviewer Figure 3A**). Sub-clusters are separated based on days post-immunization and specific genes defining day 1 sub-cluster are key antiviral and interferon-inducible genes (**Reviewer Figure 3B**). Both 3M-052 and YF-17D induced genes involved in interferon and antiviral response, such as *Irf7* and *Ifi44*, in LN pDCs at day 1 post-immunization (**Reviewer Figure 3C**). By day 28 post-immunization, some genes remained differentially expressed, notably *Irf7* and *Ifi2712a*, in 3M-052 immunized group but were undetectable in LN pDCs from YF-17D group. In addition, many of the differentially expressed genes at day 28 in YF-17D were found to be ribosomal proteins, suggesting a lack of differential response between day 28 and baseline. This has been included in the manuscript as Supplementary Fig. 5a.

Figure 3. Further analysis of LN pDCs.

(a) UMAP plot showing sub-clustering of 3M-052 pDCs. (b) Variable genes in each sub-cluster identified. (c) Differentially expressed genes in 3M-052-Alum/OVA and YF-17D group at day 1 and 28 post-immunization

3. While there is an emphasis on pathways in common between the vaccine regimens, one area not explored is differences that may underlie the different “phenotypes” of these vaccines. In particular, YF-17D induces exceptionally long-lived cellular responses, and most of the demonstrated enhancement of 3M052 has been on the humoral responses. A more in-depth comparison of differences would be useful to generate novel hypotheses would improve the manuscript.

We fully agree with the reviewer on the importance of comparing the phenotypes of the vaccines. We started by examining unique and overlapping DEGs (absolute logFC > 0.25, FDR < 0.05) at day 1 post-immunization with 3M-052 and YF-17D (**Reviewer Figure 4**). However, closer examination of the “unique” genes revealed that many had similar expression profiles after treatment with the other vaccine, but simply did not pass the significance threshold. For example, in monocytes, 3M-052 induces *Il15ra* with a LogFC of 0.60 and an adjusted pvalue of < 0.001; in YF-17D, the same gene is induced with a LogFC of 0.15 but an adjusted pvalue of 0.1. It is possible that the lack of significance in YF is simply spurious, or that a higher dose of YF vaccination is needed to significantly induce *Il15ra* in monocytes. We further speculated that a failure to pass the significance threshold may simply be due to insufficient statistical power due to a low cell count in a particular cluster. Thus, the unique genes identified in this naïve analysis might not represent true biological differences between 3M-052 and YF-17D.

Figure 4. Overlapping and unique DEGs (absolute logFC > 0.25, FDR < 0.05) at day 1 post-immunization between 3M-052 and YD-17D.

We therefore next asked how many of the DEGs identified were induced in the same direction in both vaccines. For this analysis, we took the genes that were differentially expressed (absolute logFC > 0.25, FDR < 0.05) in YF-17D, and examined what percentage of those genes were expressed in the same direction after immunization with 3M-052, regardless of FDR. For example, in monocytes, there were a total of 572 differentially expressed genes at day 1 post-immunization with YF-17D. Of those 572 genes, 538 (94%) changed in the same direction as monocytes profiled one day after immunization with 3M-052 (**Reviewer Figure 5A and 5B**). This suggests that the pattern of differential gene expression was largely similar between YF-17D and 3M-052 at an early timepoint post-immunization. Both groups potentially induced robust transcriptional changes in monocytes that could augment long-lived plasma cell differentiation, as previously suggested^{1,7}. However, some unique DEGs induced by 3M-052 in monocytes and XCR1- DCs include additional genes involved in inflammatory and IL-1b production (**Reviewer Figure 5C**).

On the contrary, we did not observe a substantial number of unique DEGs among innate cells in YF-17D group at day 1. However, some key differences we observed between both groups are the epigenetic changes in innate cells at day 28 post-immunization (Figure 5E in manuscript). Specifically, YF-17D

induces lasting chromatin accessibility of transcription factor motifs involved in NK cell activation. This suggests that the live attenuated YF-17D vaccine induces at least an imprinting in NK cells, and possibly a robust cytotoxic response, as was previously shown⁸. Future work will involve increased sampling of later timepoints, which may be able to identify more subtle or late differences in innate activation between 3M-052 and YF-17D. This analysis has been included in the manuscript as Supplementary Fig. 3.

Figure 5. Differences in gene expression between 3M-052 and YF-17D at day 1 post-immunization (A) DEGs with shared direction between 3M-052 and YF-17D at day 1 post-immunization. (B) Final number of unique DEGs in each group. (C) Enrichment of gene ontology (GO) pathways in unique 3M-052 DEGs at day 1 post-immunization.

Minor Points:

1. There are several figures where the images are too small, and the font for Greek symbols has not rendered correctly.

We have corrected this.

2. The colour code in the UMAPs is hard to follow and could be improved by denoting cells by text in figure.

We have included the text in figure.

Reviewer #2 (Remarks to the Author):

Lee, Scott and colleagues apply single-cell genomics techniques to explore the immune response in murine lymph nodes following vaccination. The study is incredibly well executed and the manuscript reads well. If there is anything that I could suggest the authors do a little more of is speculate about the mechanisms that drive the prolonged myeloid reprogramming signature at 28 days; the authors do a little bit of this but I'm curious if there could be any evidence of myeloid reprogramming in the bone marrow as others have speculated.

I recommend swift acceptance.

We thank the reviewer for the comment. It is indeed possible that 3M-052-Alum/OVA induces prolonged changes in the bone marrow since trained immunity has been shown to be sustained in the bone marrow. If anything, we noted marginal increase at day 28 in the accessibility of transcription factor motifs that are involved in myeloid differentiation in dLN monocytes, such as CEBP/a and CEBP/b. We speculate that it is thus possible for myeloid reprogramming to be maintained in the bone marrow cells in our model, and this is of interest in our future work.

References

1. Kasturi, S. P. *et al.* 3M-052, a synthetic TLR-7/8 agonist, induces durable HIV-1 envelope-specific plasma cells and humoral immunity in nonhuman primates. *Sci. Immunol.* **5**, (2020).
2. Arunachalam, P. S. *et al.* T cell-inducing vaccine durably prevents mucosal SHIV infection even with lower neutralizing antibody titers. *Nat. Med.* **26**, 932–940 (2020).
3. Arunachalam, P. S. *et al.* Systems vaccinology of the BNT162b2 mRNA vaccine in humans. *Nature* **596**, 410–416 (2021).
4. Auderset, F., Belnoue, E., Mastelic-Gavillet, B., Lambert, P. H. & Siegrist, C. A. A TLR7/8 Agonist-Including DOEPC-Based Cationic Liposome Formulation Mediates Its Adjuvanticity Through the Sustained Recruitment of Highly Activated Monocytes in a Type I IFN-Independent but NF- κ B-Dependent Manner. *Front. Immunol.* **11**, 1–13 (2020).
5. Ning, S., Pagano, J. S. & Barber, G. N. IRF7: Activation, regulation, modification and function. *Genes Immun.* **12**, 399–414 (2011).
6. Leylek, R. *et al.* Integrated Cross-Species Analysis Identifies a Conserved Transitional Dendritic Cell Population. *Cell Rep.* **29**, 3736–3750.e8 (2019).

7. Kwissa, M. *et al.* Dengue virus infection induces expansion of a CD14+CD16 + monocyte population that stimulates plasmablast differentiation. *Cell Host Microbe* **16**, 115–127 (2014).
8. Wieten, R. W. *et al.* A single 17D yellow fever vaccination provides lifelong immunity; characterization of yellow-fever-specific neutralizing antibody and T-cell responses after vaccination. *PLoS One* **11**, 1–18 (2016).

REVIEWERS' COMMENTS

Reviewer #1 (Remarks to the Author):

The authors have satisfactorily addressed all the concerns raised in the the prior review through additional analyses. I have no additional concerns and would recommend acceptance.

Reviewer #2 (Remarks to the Author):

The authors have addressed my concerns and I recommend acceptance.

Reviewer #3 (Remarks to the Author):

This is a well performed atlas-type project including some nice sc-ATAC-seq data looking at trained immunity. The authors have already revised the manuscript and addressed reviewers questions well. I had 2 additional points

1. Although Ova was added as the antigen I could not see any antigen -specific responses detailed in the figures. I appreciate this is not the main point of the paper - but on the other hand it is the main point of vaccination so just adding the basic data on B cell/antibody and T cell responses (ideally with a simple time course) would orient the reader.
2. The title is a bit general considering it is a specific set of vaccines/vaccine approaches being studied. As shown here there are likely many shared pathways activated but it probably be a fairer reflection of the content to make it more precise.

REVIEWER COMMENTS

Reviewer #3 (Remarks to the Author):

This is a well performed atlas-type project including some nice sc-ATAC-seq data looking at trained immunity. The authors have already revised the manuscript and addressed reviewers questions well. I had 2 additional points

1. Although Ova was added as the antigen I could not see any antigen -specific responses detailed in the figures. I appreciate this is not the main point of the paper - but on the other hand it is the main point of vaccination so just adding the basic data on B cell/antibody and T cell responses (ideally with a simple time course) would orient the reader.

We have provided antibody, GC B cell, and Tfh cell data in the Supplementary figure as Sup Fig. 1a and 1b.

2. The title is a bit general considering it is a specific set of vaccines/vaccine approaches being studied. As shown here there are likely many shared pathways activated but it probably be a fairer reflection of the content to make it more precise.

We thank the reviewer for the suggestion. We have revised the title.